# A Deep Learning System for Automated Quality Evaluation of Optic Disc Photographs in Neuro-Ophthalmic Disorders

**DOI:** 10.3390/diagnostics13010160

**Published:** 2023-01-03

**Authors:** Ebenezer Chan, Zhiqun Tang, Raymond P. Najjar, Arun Narayanaswamy, Kanchalika Sathianvichitr, Nancy J. Newman, Valérie Biousse, Dan Milea

**Affiliations:** 1Singapore Eye Research Institute, Singapore National Eye Centre, Singapore 169856, Singapore; 2Duke-NUS School of Medicine, Singapore 169857, Singapore; 3Department of Ophthalmology, Yong Loo Lin School of Medicine, National University of Singapore, Singapore 117597, Singapore; 4Center for Innovation & Precision Eye Health, National University of Singapore, Singapore 119077, Singapore; 5Glaucoma Department, Singapore National Eye Centre, Singapore 168751, Singapore; 6Departments of Ophthalmology and Neurology, Emory University, Atlanta, GA 30322, USA; 7Department of Ophthalmology, Rigshospitalet, University of Copenhagen, 2600 Copenhagen, Denmark; 8Department of Ophthalmology, Angers University Hospital, 49100 Angers, France; 9Neuro-Ophthalmology Department, Singapore National Eye Centre, Singapore 168751, Singapore

**Keywords:** retinal image quality assessment, artificial intelligence, deep learning, optic nerve head, papilledema

## Abstract

The quality of ocular fundus photographs can affect the accuracy of the morphologic assessment of the optic nerve head (ONH), either by humans or by deep learning systems (DLS). In order to automatically identify ONH photographs of optimal quality, we have developed, trained, and tested a DLS, using an international, multicentre, multi-ethnic dataset of 5015 ocular fundus photographs from 31 centres in 20 countries participating to the Brain and Optic Nerve Study with Artificial Intelligence (BONSAI). The reference standard in image quality was established by three experts who independently classified photographs as of “good”, “borderline”, or “poor” quality. The DLS was trained on 4208 fundus photographs and tested on an independent external dataset of 807 photographs, using a multi-class model, evaluated with a one-vs-rest classification strategy. In the external-testing dataset, the DLS could identify with excellent performance “good” quality photographs (AUC = 0.93 (95% CI, 0.91–0.95), accuracy = 91.4% (95% CI, 90.0–92.9%), sensitivity = 93.8% (95% CI, 92.5–95.2%), specificity = 75.9% (95% CI, 69.7–82.1%) and “poor” quality photographs (AUC = 1.00 (95% CI, 0.99–1.00), accuracy = 99.1% (95% CI, 98.6–99.6%), sensitivity = 81.5% (95% CI, 70.6–93.8%), specificity = 99.7% (95% CI, 99.6–100.0%). “Borderline” quality images were also accurately classified (AUC = 0.90 (95% CI, 0.88–0.93), accuracy = 90.6% (95% CI, 89.1–92.2%), sensitivity = 65.4% (95% CI, 56.6–72.9%), specificity = 93.4% (95% CI, 92.1–94.8%). The overall accuracy to distinguish among the three classes was 90.6% (95% CI, 89.1–92.1%), suggesting that this DLS could select optimal quality fundus photographs in patients with neuro-ophthalmic and neurological disorders affecting the ONH.

## 1. Introduction

Optic neuropathies cause visual loss from various pathophysiologic mechanisms, including compression/infiltration, infectious and noninfectious inflammation, ischemia, toxicity, degeneration and disorders of intraocular (glaucoma) and intracranial (papilledema) pressure. Appropriate detection of optic neuropathies is essential, requiring visual examination of the optic nerve head (ONH) at the back of the eye. The ONH (or optic disc) is the visible interface between the optic nerve and the ocular globe and can be evaluated clinically by standard ophthalmoscopy or by ocular imaging (i.e., fundus photographs). The visual appearance of the ONH is typically altered in optic neuropathies: it can be swollen at the acute stage, then evolve towards pallor or atrophy. Papilledema is a particular type of bilateral ONH swelling specifically from raised intracranial pressure, resulting from potential vision- or life-threatening lesions (brain tumours, venous sinus thrombosis, idiopathic intracranial hypertension, etc). The early detection of papilledema is essential to avoid blindness or neurological disability. Identification of ONH abnormalities (by ophthalmoscopy or on fundus photographs) can be challenging, with misdiagnosis rates by non-specialists, even when using high-quality fundus photographs, reported as high as 69% [1,2]. This high error rate can be attributed to various factors, including technical difficulties visualizing the ONH and lack of expertise in interpreting the ONH appearance, resulting in mismanagement and delayed referrals [3].

Recently, deep learning (DL) methods have been successfully used to accurately classify papilledema and other ONH conditions on standard ocular photographs [4,5,6,7]. These DL systems identify optic disc abnormalities (and in particular papilledema) with higher performance than non-expert healthcare providers, achieving an accuracy which is similar to that of expert neuro-ophthalmologists. [8]. The performance of these algorithms has been evaluated on highly curated datasets including excellent-quality photographs obtained after pupil dilation. Such performance evaluation on highly curated datasets introduces a potential bias, compared to real-life conditions, by suppressing an important intermediate processing step, the selection of non-interpretable photographs. The real prevalence of poor quality or non-interpretable ocular fundus photographs is difficult to estimate in real conditions, since pre-filtering is usually performed during image acquisition, by the camera operators, who acquire a new image, until a “good” quality image can be achieved. After this preliminary step, the prevalence of non-suitable fundus photographs remains high, typically above 10% [9,10,11,12,13], depending on multiple factors (type of camera, pupillary dilation, transparency of the ocular media, patient’s cooperation, operator’s skill, etc.). Altogether, the process of selection and suppression of non-suitable photographs is time-consuming and labour-intensive; if not performed accurately, it can cause patient inconvenience and increased costs from unnecessary referrals related to the suboptimal quality of ocular fundus photographs. 

To mitigate these shortcomings, various DL-based retinal image quality assessment systems (RIQAS) have been recently developed, in order to automatically identify high-quality photographs in common ophthalmic conditions such as diabetic retinopathy (DR) or glaucoma [14,15,16,17,18,19,20,21,22,23]. In these conditions, “poor quality” has been specifically defined, depending on the region of interest (i.e., poor identification of third-generation branches within one optic disc diameter around the macula in DR [9], or obscuration of more than 50% of the optic disc, in glaucoma [24]). These disease-specific systems are not generalizable and therefore cannot be applied to neuro-ophthalmic or neurological conditions affecting the appearance of the ONH. 

In order to address this question, we aimed to develop, train and test a deep learning system (DLS) able to automatically classify the quality of ONH fundus photographs in neuro-ophthalmic and neurological conditions, based on data from a large, international, multi-ethnic population, using multiple cameras. A DL-driven algorithm for the quality assessment of ONH images could reduce the frequency of diagnostically unusable datasets, especially in neuro-ophthalmology where data are scarce [25,26].

## 2. Materials and Methods

### 2.1. Study Design

A total of 5015 ocular photographs, retrospectively collected from 31 international neuro-ophthalmology centres in 20 countries participating in the BONSAI (Brain and Optic Nerve Study with Artificial Intelligence) Consortium [7], were used for this study. Among them, 4208 fundus photographs (including 480 optic discs with papilledema, 332 optic discs with glaucoma, 881 optic discs with other abnormalities, 2509 normal discs and 6 images with unknown diagnosis, due to no visible optic disc) were randomly selected and used for training, validation, and internal-testing. Using a standard 80/20 split-training approach, 3356 images were included in the training and validation datasets, while the internal-testing dataset contained the rest of 20% of the images (852 images). An independent, multi-ethnic external-testing dataset included 807 ocular fundus photographs collected from three expert centres in two countries (Atlanta, USA and Singapore). The external-testing dataset included 57 optic discs with papilledema, 25 optic discs with glaucoma, 146 optic discs with other abnormalities and 579 normal discs. 

The study was approved by the centralized institutional review board of SingHealth, Singapore, and by each contributing institution. The study was conducted in accordance with the principles of the Declaration of Helsinki. 

### 2.2. Image Acquisition

The study included both mydriatic and non-mydriatic fundus photographs, obtained with multiple cameras, including handheld cameras [27] (Appendix A Table A1). Of the 5015 photographs used in the training and external-testing datasets, 2663 photographs (53%) were obtained with a handheld camera. Data was collected in normal individuals and in patients with various conditions affecting the ONH photographs (i.e., papilledema and “other” ONH abnormalities including optic atrophy, optic disc drusen, optic disc swelling unrelated to raised intracranial pressure, etc.), based on robust ground truth criteria, detailed elsewhere [7]. 

### 2.3. Generation of the Quality Reference Standard

The quality reference standard (QRS) was generated from results provided post hoc by three expert clinicians who evaluated the dataset (5015 retinal photographs). Discordant labels provided by the first two graders (a fellowship-trained neuro-ophthalmologist and a senior glaucoma specialist) were subsequently adjudicated by the third grader, a senior neuro-ophthalmologist, to obtain a majority consensus. During the classification process, the three graders used the same computer, with identical screen characteristics, in identical illumination conditions. All images were labelled using the Classif-Eye semi-automated application, which facilitates visualisation and labelling of digital photographs [28]. The graders classified the images according to the following three-class QRS:

**Good quality** photographs: defined as clear retinal images, including 100% of the ONH and peripapillary area, allowing for a confident assessment of the ONH appearance.**Borderline quality** photographs: defined as those with features allowing uncertain visual assessment of the ONH health, due to suboptimal image clarity, exposure, or partial obstruction of the image.**Poor quality** photographs: defined as images not allowing an ONH evaluation, due to various limitations, such as defocus, under- or overexposure, artefacts, poorly identifiable ONH features, or partially visible ONH. Similarly, photographs that were not compatible with the images used in the training dataset (e.g., fundus autofluorescence, wide-field retinal image) were included in this category. Examples of “good”, “borderline”, and “poor” images are shown in Figure 1, Figure 2 and Figure 3. 

### 2.4. Cross-Validation

Using 5-fold cross-validation, we evaluated the generalized performance of the model. The 4208 images in the training dataset, containing 2512 “good” (60%), 1027 “borderline” (24%), and 669 “poor” quality images (16%), were divided into 5 sets, with each set distributed with 57–62% “good”, 22–26% “borderline”, and 15–17% “poor” quality images. In the 5-fold cross-validation, one unique set was chosen as a testing dataset while the remaining four sets were designated as the training dataset. The model was fitted on the training dataset and then evaluated on the external-testing dataset. This was repeated for each part of the iteration, for a total of five times, thereby reducing the risk of selection bias.

### 2.5. Image Pre-Processing and Development of Model

The 3356 images (80%) in the main dataset were used for training/validation and 852 images (20%) in the main dataset were used for the internal testing of the model. The model was then tested on an independent external-testing dataset consisting of 807 images collected from three participating centres from Singapore and Atlanta, USA. 

Our model employed the state-of-the-art EfficientNets architecture. EfficientNets possesses a scaling method optimizing the architecture of the convolutional neural network (CNN) in terms of depth, width, and resolution, compared to any other CNNs [29].

Image standardization and pre-processing were conducted before deep learning. The input images (456 × 456 pixels) were trained using EfficientNet-B5, pre-trained on ImageNet [30] images. At the last convolutional layer of the EfficientNet-B5 architecture, the feature vectors were fused into the fully connected neural network with a SoftMax layer to optimize the performance. Data augmentation which involved random horizontal rotations and cropping, adjustments to brightness and contrast, different degrees of zoom and warping as representation of real-world acquisition conditions was applied to the training dataset. The process of introducing data augmentation provides a heterogeneous distribution of the training dataset and reduces the overfitting rate during the process of deep learning [31,32].

For the training process, the QRS data was used to optimize the performance of the DLS. Cross-entropy was used as a loss function for optimizing the models. The training started with multiple iterations with a batch size of 32 images, with an initial learning rate of 0.01 and stopped at 50 epochs. For each training iteration, a stochastic gradient descent algorithm was used to optimize the loss function to train neuron weights via backpropagation; at every epoch, the performance of the CNN was assessed using the validation dataset. Subsequently, the best-predicted model from the preliminary evaluation of the internal-testing dataset was evaluated on an independent external-testing dataset.

### 2.6. Statistical Analyses

The performance of the DLS was evaluated using the one-versus-rest strategy by various performance metrics which included the area under the receiver operating characteristic curve (AUC), accuracy, sensitivity, and specificity according to our classification model (one vs. rest approach): “good” quality vs. (“borderline” and “poor”) quality, “borderline” quality vs. (“good” and “poor”) quality, and “poor” vs. (“good” and “borderline”) quality images. The overall accuracy was used to measure the performance of the model. 

Bootstrapping sampling, repeated 2000 times, was used to estimate the 95% confidence intervals (CI) of the performance metrices.

## 3. Results

### 3.1. Characteristics of Dataset

The total of 5015 fundus photographs included 3211 “good” quality photographs, 1108 “borderline” quality photographs, and 696 “poor” quality photographs. The main dataset (used for training, validation, and internal testing) included 4208 images: 2512 images with “good” quality (60%), 1027 images with “borderline” quality (24%) and 669 images (16%) with “poor” quality. The external-testing dataset (807 fundus photographs) included 699 (87%) “good” quality photographs, 81 (10%) “borderline” quality photographs and 27 (3%) “poor” quality photographs (Table 1). The distribution of the training and validation data, according to diagnosis and quality, is summarized in Table 1.

### 3.2. Grading Duration

The total average time spent by the two experts to grade the 807 photographs in the external-testing dataset was 1687 s; the same task was performed by the DLS in 9.13 s. The average time required by the two experts to grade one fundus photograph was 2.09 s. 

### 3.3. Cross-Validation

Figure 4 displays the performance of the model obtained on each cross-validation dataset. The AUCs of the testing dataset in the cross-validation range from 0.94 to 0.98 when discriminating “good” quality from (“borderline” and “poor” quality) images, 0.89–0.93 when discriminating “borderline” quality from (“good” and “poor”) images, and 0.98 when discriminating “poor” quality from the (“good” and “borderline” quality) images. The average overall accuracy of the 5-fold cross-validation was 85.0% (81.5–88.3%). 

### 3.4. Overall Classification Performance

In the internal-testing dataset, using a one-vs-rest approach, the model discriminated “good” quality vs. (“borderline” and “poor” quality) with an average AUC of 0.99 (95% CI, 0.99–1.00), an accuracy of 95.8% (95% CI, 94.8–96.8%), a sensitivity of 95.4% (95% CI, 94.0–96.8%), and specificity of 96.3% (95% CI, 94.9–97.8%) (Table 2). The model discriminated “borderline” quality vs. (“good” and “poor” quality) with an average AUC of 0.99 (95% CI, 0.99–1.00), an accuracy of 95.8% (95% CI, 94.8–96.8%), a sensitivity of 92.9% (95% CI, 90.3–95.6%), and specificity of 96.8% (95% CI, 95.8–97.9%). Lastly, the model discriminated “poor” quality vs. (“good” and “borderline” quality) with an average AUC of 1.00 (95% CI, 0.99–1.00), an accuracy of 98.8% (95% CI, 98.3–99.4%), a sensitivity of 98.4% (95% CI, 97.3–100.0%), and specificity of 98.9% (95% CI, 98.4–99.6%). The overall accuracy of the model was 95.2% (95% CI, 94.1–96.3%) in the internal-testing dataset.

In the external-testing dataset, the model discriminated “good” quality vs. (“borderline” and “poor” quality) with an AUC of 0.93 (95% CI, 0.91–0.95), the accuracy of 91.4% (95% CI, 90.0–92.9%), a sensitivity of 93.8% (95% CI, 92.5–95.2%), and specificity of 75.9% (95% CI, 69.7–82.1%) (Table 2). The model discriminated “borderline” quality vs. (“good” and “poor” quality) with an AUC of 0.90 (95% CI, 0.88–0.93), an accuracy of 90.6% (95% CI, 89.1–92.2%), a sensitivity of 65.4% (95% CI, 56.6–72.9%), and specificity of 93.4% (95% CI, 92.1–94.8%). Lastly, the model discriminated “poor” quality vs. (“good” and “borderline” quality) with an AUC of 1.00 (95% CI, 0.99–1.00), accuracy of 99.1% (95% CI, 98.6–99.6%), a sensitivity of 81.5% (95% CI, 70.6–93.8%), and specificity of 99.7% (95% CI, 99.6–100.0%). The overall accuracy of the model was 90.6% (95% CI, 89.1–92.1%) in the external-testing dataset. Figure 5 shows the confusion matrix plots and receiver operating characteristic curve (ROC) and AUC of image quality tasks on the internal-testing and external-testing datasets.

## 4. Discussion

The objective of this study was to train, develop, and test the performance of a DLS to discriminate among three quality classes of ONH photographs (“good”, “borderline”, and “poor” quality). For this purpose, we used a large number of fundus photographs, acquired from multiple international expert centres, using a large variety of desktop and handheld digital fundus cameras. The main result of this study is that this DLS could accurately classify fundus photographs as “good”, “borderline”, and “poor” quality, with an overall accuracy of 90.6% (95% CI, 89.1–92.1%). More specifically, the DLS had an excellent performance in identifying “poor” quality photographs, with an accuracy of 99.1% (95% CI, 98.6–99.6%) on the external-testing dataset. 

In order to provide a more granular view of the reality in clinics, we avoided the use of a simple, yet classic, binary classification system (i.e., “good” vs. “poor” quality photographs). Instead, we used a three-class system, including also a “borderline” quality category, hypothesizing that this class may still allow clinical interpretation of the ONH by humans, despite potential challenges posed to RIQAS [17,33]. Indeed, in a recent image quality study applied to DR, 21% of fundus photographs were deemed ungradable by the RIQAS, but a significant number were still considered as interpretable by humans [34]. Similarly, an image quality evaluation study using a binary classification system for training, achieved high performance (AUC = 100%) on an external dataset which included only images with 100% intergrader agreement. However, when “ambiguous” fundus photographs (i.e., with discordant intergrader evaluation) were added to the testing dataset, the model’s performance (i.e., ‘accept’ vs. ‘reject’) dropped to an AUC of 54.48% and 55.11%, respectively [17].

Several images in our dataset were misclassified by the DLS. However, no “good” quality image was misclassified as “poor”, suggesting that this DLS system does not misclassify, and therefore “lose”, clinically acceptable fundus photographs. The opposite was also true: no “poor” quality photographs were misclassified as “good” quality, suggesting that a large majority of low-quality photographs can be accurately identified by this DLS with the aim to be subsequently discarded prior to the analysis. Twenty-six photographs with a ground truth of “borderline” quality were misclassified by the DLS as of “good” quality, but only 2 “borderline” quality photographs were misclassified as of “poor” quality. “Borderline” quality images were defined (during the QRS evaluation by the graders) as images that are still useful for the evaluation of the ONH morphology; therefore, misclassification of a “borderline” image as a “good” quality image is not surprising and hopefully not detrimental. On the other end of the spectrum, 43 “good” quality images were misclassified as “borderline” quality, and 5 “poor” quality images were misclassified as “borderline”. Altogether, these results suggest that (1) the DLS allows for accurate identification of images of “poor” quality since only five images were included as “borderline” images and none as “good” quality; and (2) the DLS does not inappropriately reject relevant images, since the misclassification rate of “good” and “borderline” images as “poor” images is very low. A crucial follow up work that would be required is to evaluate the diagnostic performance (of humans or by a dedicated DLS) on datasets that have been pre-processed, in terms of image quality, by the presented DLS. 

Our study has a few limitations, including the relatively small number of “borderline” and “poor” photographs in the external-testing dataset (13%), although this distribution is consistent with the reality in clinics, where 8–24% of images acquired from desktop and handheld fundus cameras have been reported as ungradable [35]. Additionally, half of the photographs used in this study were obtained with mydriatic cameras; the presented results may not apply to datasets using larger proportions of nonmydriatic cameras, including wide-field cameras.

If further validated, our DLS may serve in the real world by providing immediate feedback on an image’s quality without the need to manually assess an individual image’s suitability, a process which can be time costly. Even for seasoned ophthalmologists who assess fundus photographs for diagnostic suitability daily and reflexively, it still requires a few seconds for a decision to be made. In contrast, our DLS could screen through and classify the image quality of an entire dataset of fundus photographs 187 times faster than experienced ophthalmologists. The automation of image quality screening, when applied in neuro-ophthalmology clinics, can reduce the cognitive load on both the camera operators and ophthalmologists, allowing for more focus to be spent on patient-centric care and expeditious evaluations. If further validated, such a DL system might be used in the future for screening fundus photographs in suspected ophthalmic or neurologic patients. These DL-based pre-selected patients will be subsequently referred for confirmatory, human evaluation, which can provide high liability levels.

## 5. Conclusions

A DLS can accurately evaluate the quality of ONH fundus photographs in neuro-ophthalmic conditions and could potentially function as an automated screening tool prior to the automated classification of photographs. This process can help clinicians to photographically document their fundus findings, a practice that is not yet a standard procedure in neuro-ophthalmology. Beyond documentation, the appropriate automated deep-learning-based assistance for image diagnosis will represent an opportunity for professional improvement and improved healthcare. 

Identification of poor-quality photographs by such a system (which can be embedded in a camera or available on the cloud) could facilitate higher-quality image acquisition, reducing the frequency of unusable images and improving the efficiency of image acquisition in clinics. Further studies are needed to evaluate the relative performance of humans or diagnostic DLS, when applied to the DL-based quality pre-selection of “good” and “borderline” quality photographs in neuro-ophthalmic and neurological conditions. 

## Figures and Tables

**Figure 1 diagnostics-13-00160-f001:**
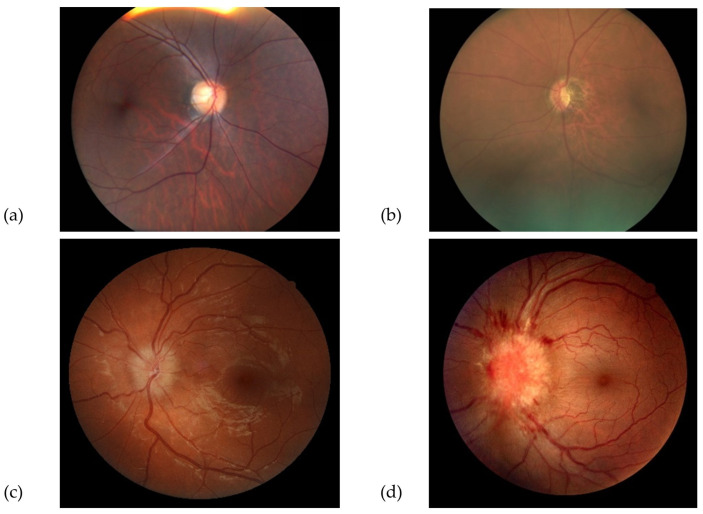
Examples of “good” quality colour fundus photographs of normal optic discs and discs with papilledema, acquired with handheld cameras (**a**,**b**) and desktop cameras (**c**,**d**).

**Figure 2 diagnostics-13-00160-f002:**
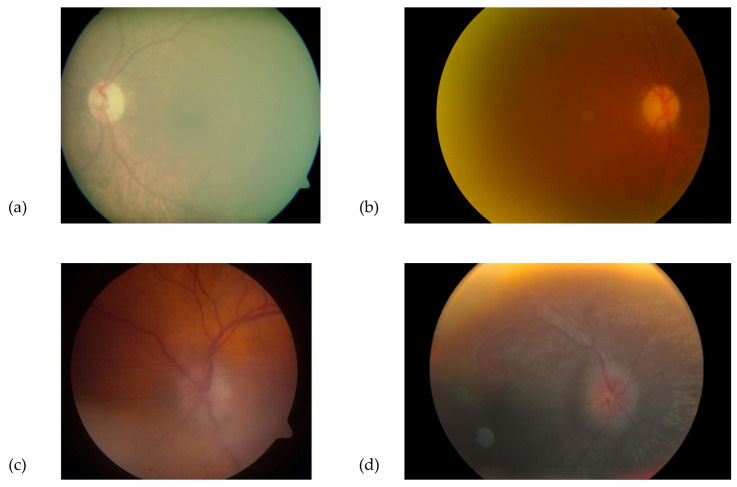
Examples of “borderline” quality colour fundus images. Despite the partial image blur, an evaluation of the optic disc is still possible (**a**–**d**)

**Figure 3 diagnostics-13-00160-f003:**
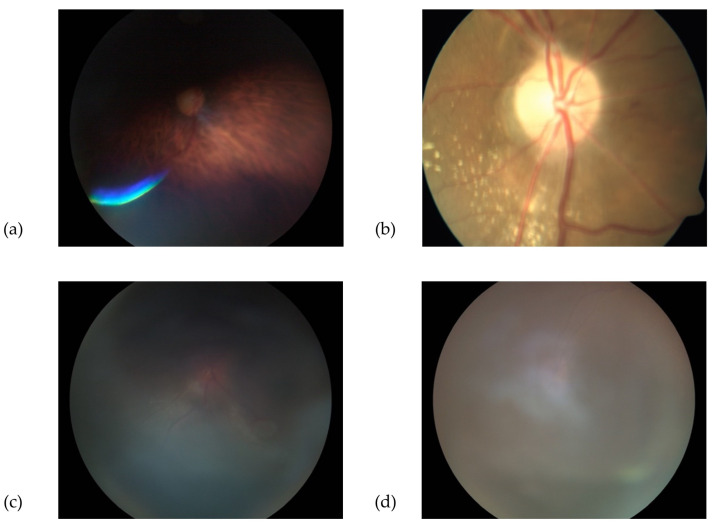
Examples of “poor” quality fundus images; the optic disc is partially masked (**a**), associated with other retinal lesions (**b**) and totally masked (**c**,**d**).

**Figure 4 diagnostics-13-00160-f004:**
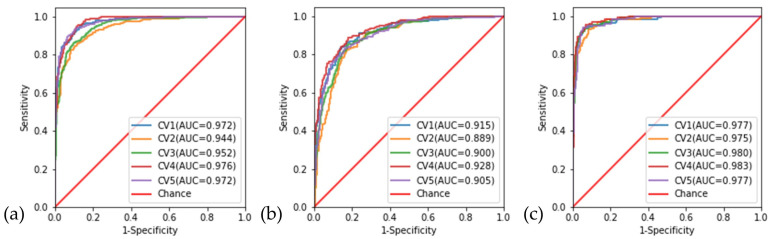
Receiver operating characteristic curves (ROC) and areas under the curves (AUC) of individual folds for the 5-fold cross-validation performed on the testing dataset. (**a**) “good” quality versus (“borderline” and “poor” quality) images; (**b**) “borderline” quality versus (“good” and “poor” quality) images and (**c**) “poor” quality versus (“good” and “borderline” quality) images.

**Figure 5 diagnostics-13-00160-f005:**
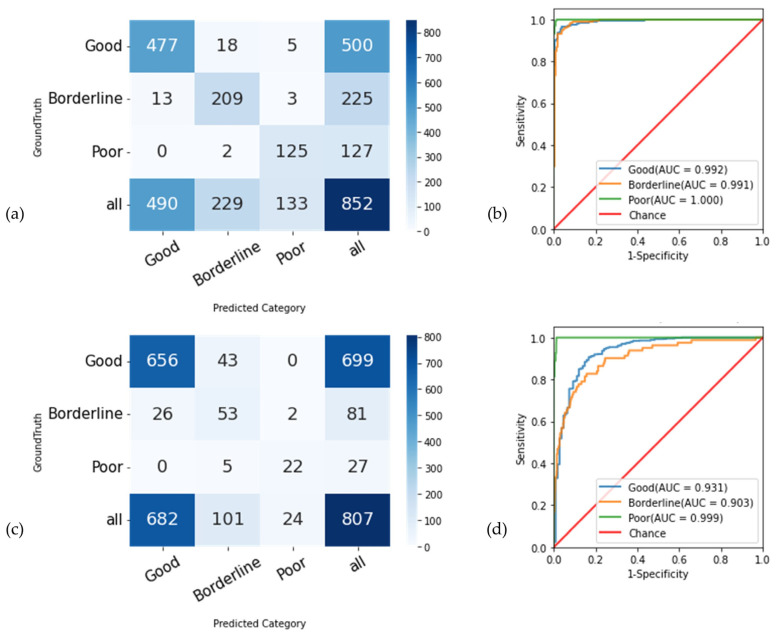
Confusion matrix plot and ROC curve of image quality task on internal-testing and external-testing datasets. (**a**): confusion matrix of the internal-testing dataset; (**b**): ROC curve of the internal-testing dataset; (**c**): confusion matrix of the external-testing dataset; (**d**): ROC curve of the external-testing dataset.

**Table 1 diagnostics-13-00160-t001:** Summary of Training, Validation, Internal-Testing and External-Testing Data Sets, According to Diagnosis of Fundus Images.

	Good	Borderline	Poor	Total
Diagnosis	number of images			
**Main dataset (training, validation,** **and internal-testing)**				
Normal Discs	1472	637	400	2509
Optic Discs with Papilledema	394	76	10	480
Optic Discs with Other Abnormalities	646	314	253	1213
Unknown Diagnosis Due to No Visible Optic Disc	-	-	6	6
**External-testing** **dataset**				
Normal Discs	488	67	24	579
Optic Discs with Papilledema	56	1	0	57
Optic Discs with Other Abnormalities	155	13	3	171

**Table 2 diagnostics-13-00160-t002:** Classification Performance of the Deep-Learning System on the Internal-testing and External-testing Dataset.

One-vs.-Rest Classification	Total	Good	Borderline	Poor	AUC (95% CI)	Sensitivity (95% CI)	Specificity (95% CI)	Accuracy (95% CI)
	No. of images					% (percentage)		
**Internal-testing dataset**								
Good vs. (Borderline + Poor)	852	500	225	127	0.99 (0.99–1.00)	95.4 (94.0–96.8)	96.3 (94.9–97.8)	95.8 (94.8–96.8)
Borderline vs. (Good + Poor)	852	500	225	127	0.99 (0.99–1.00)	92.9 (90.3–95.6)	96.8 (95.8–97.9)	95.8 (94.8–96.8)
Poor vs. (Good + Borderline)	852	500	225	127	1.00 (0.99–1.00)	98.4 (97.3–100.0)	98.9 (98.4–99.6)	98.8 (98.3–99.4)
**External-testing dataset**								
Good vs. (Borderline + Poor)	807	699	81	27	0.93 (0.91–0.95)	93.8 (92.5–95.2)	75.9 (69.7–82.1)	91.4 (90.0–92.9)
Borderline vs. (Good + Poor)	807	699	81	27	0.90 (0.88–0.93)	65.4 (56.6–72.9)	93.4 (92.1–94.8)	90.6 (89.1–92.2)
Poor vs. (Good + Borderline)	807	699	81	27	1.00 (0.99–1.00)	81.5 (70.6–93.8)	99.7 (99.6–100.0)	99.1 (98.6–99.6)

## Data Availability

Available on reasonable request.

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
