# Peer review of "A Deep Learning System for Automated Quality Evaluation of Optic Disc Photographs in Neuro-Ophthalmic Disorders"

_diagnostics, 2023, doi:10.3390/diagnostics13010160_

Round 1

Reviewer 1 Report

I will not comment on technical aspects of the manuscript (photographic technique, AI/DL algorithm, statistical approach and so on) since I do not have the necessary expertise.

Rather, my comment will involve some general issues.

Technologies employed in diagnostic process of eye diseases are here to stay. When a new technology or – as in the case of this manuscript - a new procedure is tested, other than concluding that (line 321-3) “a DLS can accurately evaluate the quality of ONH fundus photographs in neuro- ophthalmic conditions, without human intervention, and could potentially function as an automated screening tool prior to automated classification of photographs”, ther should be room for additional questions such as: “will it help me to be a better ophthalmologist (or neuroophthalmologist)? or “will the patients benefit significantly from the introduction of this procedure in daily clinical practice”?

My main concern is “without human intervention”. Is it mainly a matter of time sparing? Are machines really better than humans in detecting optic nerve diseases? At lines 52-to-57 authors say that “identification of ONH abnormalities (by ophthalmoscopy or on fundus photographs) can be challenging, with misdiagnosis rates by non-specialists, even when using high quality fundus photographs, reported as high as 69%. This high error rate can be attributed to various factors, including technical difficulties visualizing the ONH and lack of expertise interpreting the ONH appearance, resulting in mismanagement and delayed referrals.” This assumption is depressing and represents a crushing defeat since I am an old hopeless romantic ophthalmologist who still firmly believes in human mastery over machines, or if you prefer in low-tech over hi-tech (am I a boomer?). Clinical experience acquired over time with patience and dedication, a good slit-lamp, a 90D (or a 60D) hand-held Volk lens are still unparalleled for optic disc evaluation.

In conclusion, I ask the Editor to accept the manuscript as it stands. Technically it has all the requirements requested for being accepted on a special issue. Maybe a few lines could be added by the authors, if considered worthwhile, something about how much charming is for doctor eyes to have the privilege and the liability to evaluate the fundus of a patient eye.

Reviewer 2 Report

This manuscript describes the development and accuracy of a deep learning model to automatically assess the quality of fundus photos of neuro-ophthalmic conditions. The author group (BONSAI) has previously published the accuracy of a deep learning model to distinguish between pathologic and normal optic nerves. However, in the prior study, good quality photos were selected by humans. In order to implement the BONSAI model in settings without experts to judge whether the photos are appropriate for AI classification, an automated method of judging image quality is necessary. The authors applied similar AI techniques and showed that their model had high accuracy in classifying image quality, especially when selecting poor quality images not suitable for automated grading. The methods and conclusions are sound. A few minor comments are below.

Introduction

Line 80 – why are deep learning algorithms to assess image quality of fundus photos for glaucoma assessment not applicable to neuro-ophthalmology? They are both focusing on the optic nerve.

Methods

Line 96 – if the authors believe that the criteria for good quality images for glaucoma is different from neuro-ophthalmic disorders, why are fundus photos of glaucoma for training, validation, and internal and external testing?

Supplemental table A1. It looks like the authors did not include Optos/widefield camera images. This should be explicitly stated in the methods, as the accuracy of the fundus photo classification may be different when using widefield photos.

Results

Is it possible to a subgroup analysis of different types of cameras to see if the model’s accuracy is different for mydriatic vs. non-mydriatic photos? This may determine whether the DLS is more appropriate for certain cameras.

Discussion – the authors correctly point out that a limitation of the study is the small number of poor photographs in the external testing dataset. I would further add that the external testing dataset had only 3 photos of pathologic optic nerves that were poor quality. The highest risk clinical scenario is for a poor quality photo of an abnormal optic nerve to undergo classification with an erroneous result, as a serious neurologic disorder could be missed. It would be helpful to know if the 3 poor quality photos of abnormal optic discs were correctly graded as poor by the DLS.
